# Group-Based Physical Activity Interventions Targeting Enjoyment in Older Adults: A Systematic Review

**Rachel M. Creighton** [1,*] **, Kyle F. Paradis** [2] **, Nicole E. Blackburn** [1] **and Mark A. Tully** [3]

1 School of Health Sciences, Institute of Nursing and Health Research, Faculty of Life and Health Sciences, Ulster University, Jordanstown BT37 0QB, UK; ne.blackburn@ulster.ac.uk
2 School of Sport, Faculty of Life and Health Sciences, Ulster University, Jordanstown BT37 0QB, UK; k.paradis@ulster.ac.uk
3 School of Medicine, Faculty of Life and Health Sciences, Ulster University, Londonderry BT48 7JL, UK; m.tully@ulster.ac.uk
* Correspondence: creighton-r3@ulster.ac.uk

**Abstract:** In previous research, older adults reported they preferred enjoyable exercise programmes. The aim of this systematic review is to identify the components of enjoyable group-based physical activity interventions for older adults. Eleven electronic databases were searched in May 2021. Inclusion criteria were: Community-dwelling, ≥60 years of age, group-based physical activity, controlled intervention studies, designed to promote enjoyment, and included a measure of enjoyment or physical activity. Characteristics of the studies, intervention content, and enjoyment outcomes were extracted. Six studies, involving 1205 participants (Mean = 72.9 years old, 71.3% female, 67% European countries) were included. While enjoyment outcomes ($n$ = 3) appeared to improve, there was wide heterogeneity in measurement tools, making it difficult to compare studies. Enjoyable group-based physical activity for older adults may consist of a supportive instructor and peers, creating a shared positive experience. Components included confidence building through competence, and courage experienced in the company of others. Within a physically supportive environment, older adults have the potential to generate social support to enjoy being physically active together. Future studies should include measures of enjoyment and device-based physical activity. Furthermore, a standardised definition of enjoyable physical activity for older adults is needed to inform the design of future interventions.

**Keywords:** physical activity; enjoyment; older adults

## 1. Introduction

### 1.1. Physical Activity for Older Adults

Physical activity benefits physical, cognitive, and social functioning [1], and positively impacts healthy ageing [2]. Most older adults do not achieve the recommendations of multi-component physical activity on at least three or more days a week, at moderate or greater intensity [3,4]. Engaging with the social motivators of physical activity in older adults may be more effective than focusing on the health benefits of physical activity maintenance as a motivator [5]. To that end, a recent definition of physical activity recognises it is " . . . influenced by a unique array of interests, emotions, ideas, instructions, and relationships" [6], yet the social environment is often overlooked in intervention design [7].

### 1.2. Enjoyment as a Predictor of Physical Activity

The affective variables that predict physical activity behaviour include enjoyment, positive attitude, and intrinsic motivation [7]. In addition, social support for physical activity has been found to have a role in enhancing positive affect [8]. Creating experiences that may positively impact and motivate future physical activity behaviours should be an important consideration in the design of interventions.

A conceptional framework of affective determinants of physical activity [9] can be used to inform how enjoyment may be experienced across a physical activity programme: (1) affective response—"feel good" during or immediately after physical activity; (2) affect processing—automatic and reflective processing of previous affective responses; (3) affectively charged and reflective motivational states—elicited through, for example, intrinsic motivation and intention; and (4) other environmental and cognitive factors, including the physical and social context in which the activity is taking place. Therefore, how the exercise experience is remembered may be influenced by other social factors, aside from the physical exertion.

### 1.3. Behaviour Change and Maintenance

Behaviour Change Techniques (BCTs) are observable and replicable active components in intervention content designed to change behaviour [10]. Components that enhance social support and increase the enjoyment of physical activity have been found to promote engagement in physical activity among older adults [11]. The use of an established behaviour change theory is also recommended for physical activity interventions to give more effective and consistent results [1].

Research suggests the benefits of enjoyable physical activity interventions on maintenance. Enjoyment correlates with physical activity maintenance among older adults [12,13]. A meta-synthesis of qualitative reviews demonstrated that physical activity was more acceptable among older adults when it was perceived as enjoyable and social [8,14]. However, the subjective nature of "enjoyment" is poorly defined in older adults' physical activity literature. In youth sport, Wiersma [15] identified that it is often used interchangeably with interest, fun, liking, and intrinsic motivation. Previous systematic reviews have not focused on enjoyment as part of the design, or as a primary outcome, for older adults. There is therefore a need for a systematic review to synthesise current knowledge on enjoyable physical activity group-based interventions for older adults, to inform the design of future interventions.

### 1.4. Aim of This Review

The aim of this systematic review, therefore, is to identify the components of physical activity interventions for older adults that target enjoyment as part of their design or have measured enjoyment-related constructs as an outcome or consequence. The objectives are to: (1) describe the characteristics of interventions, mapping to affect typology, BCTs, and behaviour change theories; and (2) investigate the effects of the interventions on measures of physical activity and enjoyment.

## 2. Materials and Methods

This systematic review was conducted following the guideline of PRISMA 2020 and has been registered on PROSPERO (registration number is CRD42021250906).

### 2.1. Literature Search Strategy

Eleven electronic databases were searched on 7 May 2021: CINAHL (EBSCOHost); SPORTDiscus (EBSCOHost); AMED (OVID); EMBASE (OVID); PsycINFO (OVID); MEDLINE (OVID); Scopus; The Cochrane Central Register of Controlled Trials (CENTRAL); Web of Science (Clarivate); OpenGrey; and ProQuest Dissertations & Theses A&I.

Subject headings and key words (title and abstract) were used by authors RMC and MAT to develop the search strategy in MEDLINE (OVID) and adapted for other databases in collaboration with a Subject Librarian. Search terms included those relating to "older adult" and "group-based" and "physical activity" and "affect", and are detailed in Table S1 (Table S1: Ovid Medline Search Strategy). There were no search limitation filters and all publication dates were included until 7 May 2021.

### 2.2. Inclusion and Exclusion Criteria

The included studies had to involve community-dwelling adults aged 60 years and older. All types of physical activity interventions that were conducted in part or entirely in a group setting of at least four weeks duration were included, regardless of whether they were conducted in a clinical or non-clinical setting. Intervention studies of randomised controlled trials (RCTs), observation studies, and controlled trials—quasi-experimental (including non-randomised) were included, if they had a control arm for comparison. The comparison could be a standard exercise intervention (e.g., exercise referral programme), minimal intervention (e.g., a booklet with information about the benefits of activity), or no intervention (e.g., waiting-list control group). Included interventions had to either specifically state they targeted enjoyment in their design or have an enjoyment-related outcome measure (quantitative or qualitative), even if it was not the primary outcome. If the study was designed to be enjoyable but there was no measure of enjoyment, a physical activity outcome measure was required for that study to be eligible. All years and places of publication were included.

Studies were excluded if the mean age of participants was under 60 years, or if participants lived in nursing or residential institutions. Interventions that did not involve group-based physical activity and those with less than four weeks duration were excluded. Any study without a control group was excluded. Studies that did not report any physical activity or enjoyment-related outcome measures were excluded. Conference abstracts and studies that were not published in English were excluded.

### 2.3. Screening

References identified in the searches were imported to Endnote 20.1 reference management tool (Clarivate Analytics, Philadelphia, PA, USA) and duplicates removed. The remaining references were exported to Covidence systematic review software 1.0 (Veritas Health Innovation, Melbourne, VIC, Australia) for further removal of any remaining duplicates and screening. Titles and abstracts of the remaining references were independently screened against the inclusion criteria by two reviewers (RMC and MAT). Following this, the full text of all remaining articles was independently checked for inclusion by the same reviewers, before making a final decision on eligibility. The reference lists of included articles were also hand searched for potentially eligible articles. Any discrepancies were discussed with a third reviewer (KFP) until a consensus was achieved. The reason for excluding studies was documented.

### 2.4. Data Extraction

Data was extracted and recorded in Covidence systematic review software 1.0 (Veritas Health Innovation, Melbourne, VIC, Australia) by RMC and independently reviewed by MAT. The authors were contacted for additional information as necessary. Discrepancies were finalised by consensus with a third reviewer (KFP). The data extraction template on Covidence systematic review software was amended to include further information on the study identification, methods, population, intervention, and outcomes.

Details of intervention outcomes relating to enjoyment and physical activity were extracted. In addition, types of affect, BCTs, and behaviour change theories were extracted and independently reviewed. Affect was categorised as (a) affective response, (b) affect processing, (c) affectively charged motivation, and (d) other environmental and cognitive factors. BCTs were mapped according to the 93 hierarchically clustered techniques [16] and independently reviewed (RMC and MAT). Information regarding the behaviour change theories documented in the study to underpin the design of the intervention were extracted. Finally, individual, interpersonal, social, and environmental factors that may have influenced or interacted with the perception of enjoyment within the group were noted.

## 2.5. Risk of Bias Assessment

Risk of bias was independently assessed by two reviewers (RMC and MAT) and finalised by consensus of a third reviewer (KFP). For individually randomised, parallel-group trials, the Revised Cochrane Risk of Bias Tool for RCTs (RoB2) was used [17]. For non-randomised controlled studies, the Risk of Bias in Non-randomised Studies of Interventions (ROBINS-I) was used [18]. Risk of bias for each study was categorised as low if all the criteria were assessed as having a low risk of bias, as high if one or more of the criteria was assessed as high, or unclear if no criteria were assessed as high, but one or more of the criteria was assessed as unclear.

The certainty of evidence of each included study was considered, according to the Grading of Recommendations, Assessment, Development, and Evaluations (GRADE) systematic framework [19]. However, due to the limited number of included studies and the heterogeneity in design and outcome measures, the authors agreed that it was inappropriate to assign a level of certainty for each outcome.

## 2.6. Data Analysis

Continuous data on enjoyment, measured using a questionnaire, were extracted as the overall score or a score of a sub-domain. Physical activity was also measured as a continuous (e.g., minutes of physical activity per day and adherence data) variable. Any enjoyment and physical activity outcomes in the intervention group, compared to the control group, were reported.

Due to the high heterogeneity between studies (design, intervention, and outcome), a meta-analysis was not appropriate. Data were therefore analysed narratively and reported according to synthesis without meta-analysis (SWiM) reporting guidelines [20].

## 3. Results

### 3.1. Study Selection

A total of 20,998 citations were obtained from 11 databases after applying the pre-defined search strategy (Figure 1). After removing duplicates, the titles and abstracts of 10,012 papers were screened, of which 223 papers were retained for full-text screening. Six interventions (from nine articles, where four articles related to the same intervention) were eligible for inclusion. The GrOup-based physical Activity for oLder adults (GOAL) study was referred to in four papers [21–24] containing both enjoyment outcomes (affective attitudes from a GOAL sub-sample provided by the author on request) [23] and physical activity adherence outcomes [22]. No additional studies identified from the reference lists of included articles were eligible.

### 3.2. Study Characteristics

Among the six studies included (Table 1), the publication dates ranged from 2005 to 2021. Four studies were conducted in Europe (67%), one in North America (17%), and one in Asia (17%). The participants' mean (SD) age was 72.9 (5.5) years. Sample sizes ranged from 37 to 627, with the mean (SD) sample size of 200.3 (228.5), and were mainly composed of females (71.3%), including one female-only study [25]. Half of the trials employed randomisation, including a feasibility and a pilot trial. The control groups varied across exercise interventions (home based, standard exercise intervention not exclusive to older adults), to non-exercise interventions (social programmes), to no intervention (healthy younger adults or waiting list control).

Two studies recruited inactive participants [25,26] and, although baseline physical activity levels were not consistently reported, two studies indicated participants had moderate mobility [25,27], including participants with a high risk of falls [27]. Two studies reported a good health status [22,28]. Where reported, baseline physical activity levels were sub-optimal [22,26,28,29], including one study with a range of non-frail to mild frailty levels. However, some control groups achieved physical activity recommendations [26], which included younger participants [22,28].

Enjoyment was explicitly targeted in the design of two studies through tea-breaks and celebrations of achievements [26,27], however these studies did not report enjoyment as an outcome measure. The relationship between enjoyment and adherence was implicitly targeted in four studies. These studies compared the outcomes of groups according to different frequencies of meeting, age, gender identities, and virtual formats. In investigating the relationship between physical activity engagement and mood, enjoyment was qualitatively measured at the end or following an intervention [28], and indirectly as a positive engagement subscale of an exercise-induced feeling inventory pre and post intervention [25]. In investigating the relationship between enjoyment in physical activity and adherence, enjoyment was measured directly using the Physical Activity Enjoyment Scale (PACES) pre and post intervention [29], and an affective attitudes towards physical activity scale was used during and post intervention [23].

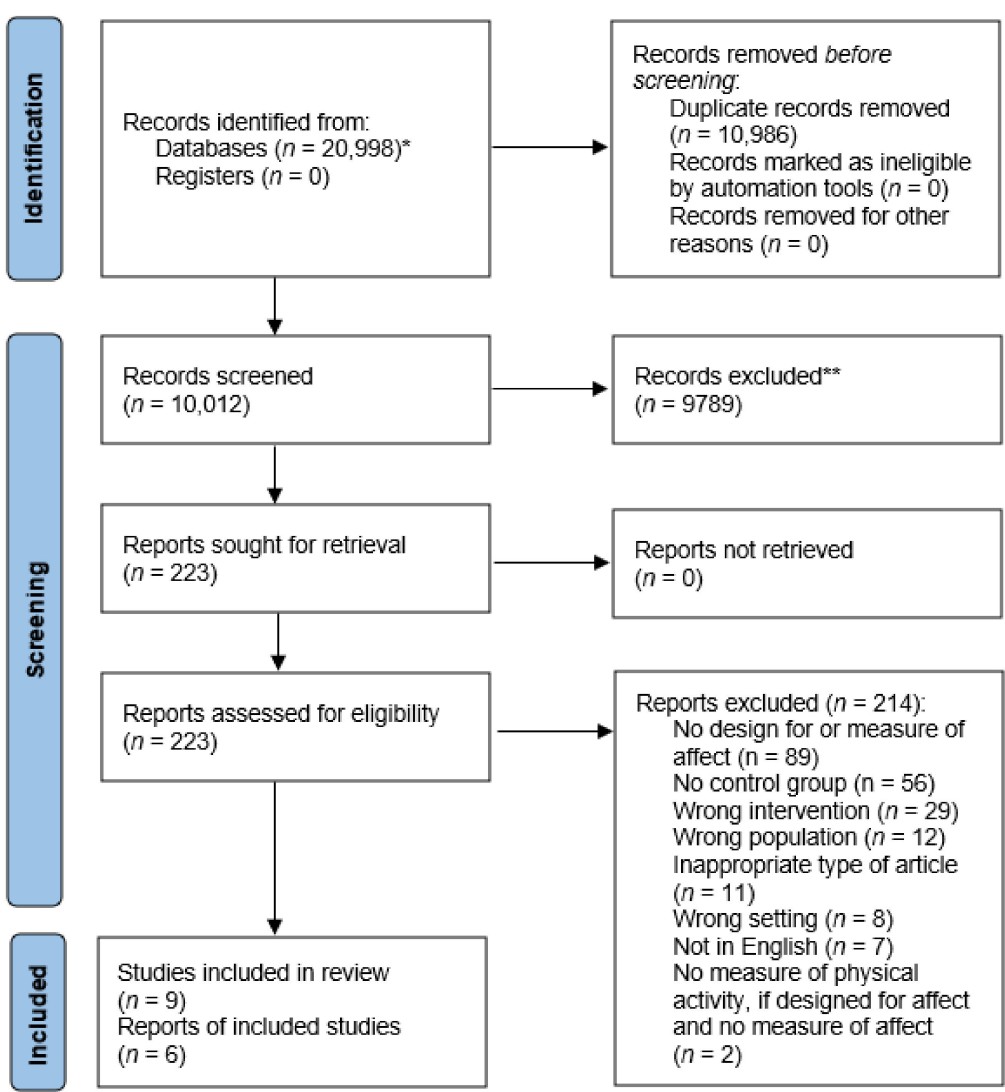

**Figure 1.** PRISMA flow diagram. * The search results from individual databases are recorded in the Supplementary Materials (Figure S1). ** No automation tools were used; all records were excluded by a human.

**Table 1.** Characteristics of included studies.

| Author (Year) | Country | Age (Years) Mean (SD) | Sample Size | Gender (% Female) | Study Design | Control Group | Participants' Characteristics | Enjoyment Aspect of Design | Enjoyment Related Outcome Measures | Measurement Timepoints |
|---|---|---|---|---|---|---|---|---|---|---|
| Baez, et al., (2017) [29] | Italy | 70.9 (5.7) | Total: 37; I = 20; C = 17 | 73% | Randomised pilot trial | Standard individual home-based training program | Ranging from non-frail to mild frailty level | Investigating the relationship between enjoyment in physical activity and adherence | PACES: 16 items (5 points per item) | Week 0 Week 10 |
| Beauchamp, et al., (2018, 2021) [22,23] | Canada | 71.57 (5.4) | Total: 627; I$^{(SASG)}$ = 236; I$^{(SAMG)}$ = 210; C$^{(MAMG)}$ = 181 | 71% | Randomised controlled trial | Standard exercise group composed of mixed age and gender | Health status mainly good–very good, married (50.6%), White (82.1%), retired (79.1%) | Affective attitudes towards physical activity predicted as mediators of adherence, in grouping similar identities together | Affective attitudes towards physical activity: three semantic differential items (7 points per item) | Week 12 Week 24 |
| Fox, et al., (2007) [28] | Italy, France, England | 75.6 (3.9) | Total: 176; I = 112; C = 64 | 56% | Non-randomised controlled trial | No intervention: healthy younger adults aged 20–37 years (n = 45) | Majority with no serious conditions of ill health, at least secondary-level educated, married | Investigating the relationship between engagement in physical activity and mood | Enjoyment as a qualitative theme on the effect on participants well-being | Week 52 [a] 8–12 months post-intervention [b] |
| Inokuchi, et al., (2007) [27] | Japan | 80.8 (6.5) | Total: 268; I = 144; C = 124 | 84% | Multi-centre non-randomised controlled trial | Weekly social programmes in day centre | Criteria included 5 or more risk factors for falls | Recreational tea breaks were explicit in the design to encourage friendly relations | NR | NR |
| Matsouka, et al., (2005) [25] | Greece | 64.8 (4.7) | Total 55; I = 45; C = 10 | 100% | Non-randomised controlled trial | Non-exercisers (members of the Public Care Institutes for the Elderly) | Inactive female-exclusive criteria, members of Public Care Institutes for the Elderly (PCIE) | Groups were designed to meet 0, 1, 2, and 3 times per week, examining the effect of group exercise frequency on mood | Exercise-induced feeling inventory, positive engagement subscale: 12 items (5 points per item) | Week 0 Week 12 |
| Stathi, et al., (2020) [26] | UK | 73.8 (6.9) | Total: 39; EG = 22; CG = 17 | 44% | Feasibility randomised controlled trial | Waiting-list control group | Sedentary, retired, inactive [c], capable of walking at least 200 m, and no diagnosis of dementia | Designed to support socially disengaged and inactive older adults to get out of the house and celebrate achievements | NR | NR |

[a] MMC site in England; [b] KC site in England; [c] Reported less than 20 min per week in the past month in MVPA. Abbreviations: C = control group; I = intervention group; IPAQ = International Physical Activity Questionnaire; MAMG = mixed age, mixed gender; MVPA = moderate to vigorous intensity physical activity; NR = not reported; PA = physical activity; PACES = Physical Activity Enjoyment Scale; SAMG = similar age, mixed gender; SASG = similar age, same gender; SD = standard deviation.

*3.3. Intervention Descriptions*

Intervention descriptions are reported according to an adapted TIDieR framework (Table 2). Most interventions described multi-component exercise programmes, including strength, balance, coordination, and flexibility, as well as aerobic moderate intensity exercise. This is consistent with the falls prevention [27,29] and active ageing [21,25,26,28] basis of programmes. Each programme also provided opportunities for social interactions, during, following, or in addition to the group exercise session.

Three of the studies used trained exercise professionals as facilitators, two studies used peer facilitators, and one study used public health nurses to deliver the programme. Five of the six interventions were face-to-face group exercise programmes, including one that connected participants with existing local programmes, and one was virtual. Three studies adapted previously used interventions to include females [21], to be delivered by public health nurses [27] and to be delivered online [29].

Most interventions occurred in community settings (of which one was age-specific [25]), and ranged in length from 8 to 52 weeks (Mean (*M*) = 23). The eight-week intervention refers to a ten-week programme, which included pre-course technology training that was non-group based and therefore not incorporated in intervention weeks [29], and likewise the 11-week intervention was part of a 12-week study, including measurements [25]. Excluding one study that did not provide data for meeting frequency and session duration [26], interventions met one to three times per week (*M* = 2), for a duration of 30–120 min (*M* = 66 min). The overall number of sessions ranged from at least 7 to 104 sessions (*M* = 40), equating to an average of two hours of planned group-based activities per week with additional home-based exercises encouraged.

Five studies measured adherence to the group-based exercise intervention (Table 2) by the number of classes attended (computer log in = 1; swipe card = 1; class register = 3). Attendance ranged from 45.1 [22] to 93% [28]. The study that investigated the frequency of weekly sessions did not however have a measure of adherence [25]. Adherence outcome measures are discussed further in Section Adherence Outcome Measures.

### 3.3.1. Behaviour Change Techniques

The intervention description of each study was mapped to specific BCTs. Out of 93 potential BCTs, 19 (20.4%) were identified, with each study using a range of one to nine techniques (*M* = 4.7). The common groupings of BCTs were antecedents (*n* = 6), social support (*n* = 5), goals and planning (*n* = 4), and shaping knowledge (*n* = 3). The most used techniques were restructuring the social environment, adding objects to the environment, social support (unspecified), instructions on how to perform the behaviour (*n* = 3), and demonstration of the behaviour (*n* = 2). Examples of the use of BCTs used in the included studies are displayed in the Supplementary Materials (Table S2).

### 3.3.2. Behaviour Change Theory

Two interventions (33.3%) reported an underpinning behavioural change theory. These were primarily the self-determination theory (SDT) [30], utilised in the peer volunteering active ageing intervention, Active, Connected, Engaged (ACE) [26], and self-categorization theory [31], which underpinned the GOAL trial [21].

*3.4. Results of Individual Studies*

3.4.1. Quantitative Findings

Enjoyment Outcome Measures

Enjoyment increased in all intervention groups. The control groups also increased [29] or maintained their enjoyment score [25], except for one study that used a mixed-age control exercise group [23]. For Beauchamp, et al., (2021), the affective attitudes were higher in the mixed-gender group (5.97 (1.19)), compared to the same-gender group (5.86 (1.54)) of older adults, compared to the mixed-age, mixed-gender control (5.79 (1.04)).

**Table 2.** Content of group-based physical activity interventions targeting enjoyment in older adults.

| Author (Year) | Physical Aspect of Intervention | Social Aspect of Intervention | Basis of Exercise Program | Who Delivered the Intervention | Mode of Delivery | Setting | Length (Weeks) | No. Session per Week | Session Duration (Mins) | How Adherence Was Assessed | Adherence | Theory |
|---|---|---|---|---|---|---|---|---|---|---|---|---|
| Baez, et al. (2017) [29] | Muscle strengthening and balance retraining exercises | Virtual gym classes with messaging and persuasion features | OTAGO falls prevention programme | Training coach | Gym central trainee mobile app | Online | 8 | 2+ | 30–40 | 1. % of sessions attended 2. % length of exercise videos watched | 1. 85% sessions attended 2. Mean (SD) 91.75% (12.46) | NR |
| Beauchamp, et al., (2018, 2021) [22,23] | Moderate-intensity exercises | Exerciser t-shirts and post-workout gatherings, delivered by older adults | GOAL trial informed by Lively Lads case study | Peer facilitators | Face to face | YMCA centre | 24 | 3 | 50–60 | Electronic attendance records from access cards | Week 12: 52.2% [a] Week 24: 45.1% [a] | Self-categorization theory, social cognitive theory, theory of planned behaviour |
| Fox, et al., (2007) [28] | Aerobic exercise, machine-based strength training, Tai Chi, and flexibility exercise | Initial intensive support and social opportunities | Better Ageing Project | Researchers and exercise professionals | Face to face | University Campus Health and Wellness Clubs | 52 | Centre = 2; Home = 1 | 60–90 | Class register and home log | 93% attendance at exercise classes and 85% for home exercises | NR |
| Inokuchi, et al., (2007) [27] | Stretching and strengthening exercises against gravity and balance retraining exercises included stepping, tandem walking, and sideways walking | Recreational and tea breaks | Nurse-led falls prevention exercise programme | Nurse-led | Face to face | Community centre | 17 | Centre = 1 (supplemented by daily home exercises) | 120 | Class register and home log | 90.9% attendance at classes Home exercises: Mean = 4.3 days a week | NR |
| Matsouka, et al., (2005) [25] | Flexibility, strength, coordination (outdoor and indoor leisure activities, callisthenic exercises), aerobic (walking and aerobic exercises), flexibility, balance, agility, muscular coordination (games and recreational activities) | Focus on leisure and recreational activities | Training programme based on "Long-term Physical Activity Workshop" for isolated older adults | Trained exercise specialist | Face to face | Public Care Institutes for the Elderly | 11 | 1–3 | 45 | NR | NR | NR |
| Stathi, et al., (2020) [26] | Promotion of a range of local physical and non-physical activity opportunities | Initial individual peer support and joint visits, two group social events | Active, Connected, Engaged (ACE) programme | Peer led by trained activators | Introduction to community face-to-face groups | Existing community groups | 24 | NR | NR | % of sessions with their activator attended | 100% | Self-determination theory |

[a] SASG and SAMG intervention groups combined. Abbreviations: NR = not reported; SASG = similar age, same gender; SAMG = similar age, mixed gender.

Physical Activity Outcome Measures

One study directly measured the physical activity level. This was a feasibility trial not powered to detect the statistical significance. Moderate to vigorous intensity physical activity (MVPA) was measured using an accelerometer [26] on an inactive sample. MVPA improvement was lower in the intervention group (1-min-per-day difference versus 5.6 min/day in control group) [26]. However, the control group had higher baseline physical activity levels (34.5 min/day versus 17.7 min/day) than the intervention, which included non-physical social activities, and participants did not comply well with wear time of the waist-worn accelerometers.

Adherence Outcome Measures

Among the five studies that measured adherence (Table 2), the lowest levels were reported by Beauchamp, et al., (2018). Nonetheless, it was found that older adults randomised to the mixed-gender group ($p < 0.001$) participated in significantly more exercise classes compared to the mixed-age, mixed-gender group at both 12 and 24 weeks. There was no significant difference between older adults randomised to the same- or mixed-gender groups at 12 ($p = 0.074$) or 24 weeks ($p = 0.187$) [22].

In contrast, Baez, et al., (2017) found high levels of adherence in both groups, however their results suggested that a higher fitness level in the control group affected adherence, whereas the initial level of fitness in the intervention group did not appear to affect adherence in the online groups [29]. Inokuchi, et al., (2007) achieved 90.9% attendance in the intervention group and also 91.2% in the control group (social programme in a day centre that elderly people already utilised once a week). The intervention in this non-randomised study was popular in the context of Japan, where exercise classes are encouraged at government level for the benefit of older adults maintaining their physical function.

Fox, et al., (2007) reported 93% attendance in group and 85% in home-based element of the intervention, but there was no comparison control measure [28]. Although Stathi, et al., (2020) did not report overall attendance, or attendance at the group social events, 100% of those who completed the intervention, achieved the minimum level planned, of engaging with their activator at least seven times (two one-to-one initial meetings in weeks 0–2: motivation stage, three joint visits in months 1–3: action stage, and two joint visits in months 3–6: maintenance stage). The two social events were reported to be well received and attended and recommended for future research into long-term maintenance of positive outcomes [26].

3.4.2. Qualitative Findings

Types of affect [9] were mapped for each intervention to illustrate how enjoyment was described, experienced, or implied. Four themes were identified. The reoccurring theme across all affect types was courage. The embodiment of courage as giving and receiving encouragement also impacted the physical and social contexts. The affective response to exercise included supporting, interacting, caring, and laughing during exercise [23,26,28]. These social connections were affectively processed following exercise, where the shared experiences and company was remembered as enjoyable [23,29].

A sense of achievement, returning to companions, gaining confidence, and knowledge affectively charges motivation in anticipation of exercise [29]. A supportive environment enabled opportunities for participants to interact [23,27–29]. Courage taken from the group leader encourages individual confidence to participate and courage exchanged within the group encourages a sense of community [26].

A sense of challenge, competence, and courage all appear inter-related in the affective response during exercise. For example, strength training provided a challenge, as it involved gradually increasing the level of exertion and taking the opportunity to increase in ability levels [29]. Competencies, such as monitoring heart rate, may have led to improved confidence [25] and courage to exercise safely. The safety aspect was referred to in qualitative results as a contributing factor to a supportive environment [28], and alludes to

creating the conditions for challenge and courage, with reassurance of progress, alongside the physical competence to cope with exercise loading.

Specific to the affect processing following exercise is the perception of company, perceiving the group to be similar in physical abilities, interests [23], and needs [28]. This may be experienced in the exercises, in social meetings, or at an environmental level where the leader is of a similar age and the place in which the programme is conducted [26]. The positive social interactions, where everyone is welcomed, and has the opportunity to get to know each other, in this social, supportive community confirms the group identity over time.

Specific to affectively charged motivation, anticipating future exercise, is confidence with courage, feeling a sense of worth and satisfaction [23], achievement, and valuing the effort to counteract the ageing process [28]. The reinforcement of self-confidence and self-esteem through certain exercises may relate to affectively charged motivation in anticipating future exercise [25].

*3.5. Risk of Bias*

Two randomised studies (Figures 2 and 3) were assessed as having a low risk of bias [22,23,29], with the third rated as an high risk [26], due to the uncertainty about random-sequence generation, deviations from the intended intervention, and high risk of missing outcome data. Non-randomised studies (Figures 4 and 5) were mainly a low risk of bias, however information bias for each study was unclear [25,27,28].

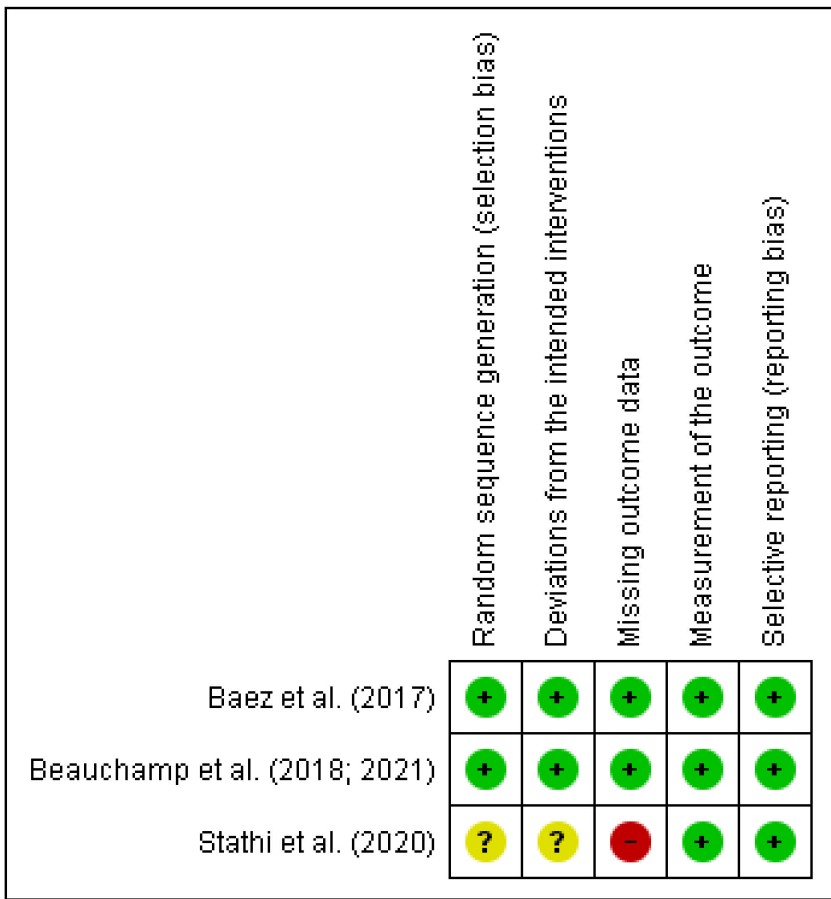

**Figure 2.** Risk of bias summary for randomised studies: review authors' judgements about each risk of bias item for each included study. (Baez, et al., (2017) [29]; Beauchamp, et al., (2018; 2021) [22,23]; Stathi, et al., (2020) [26].

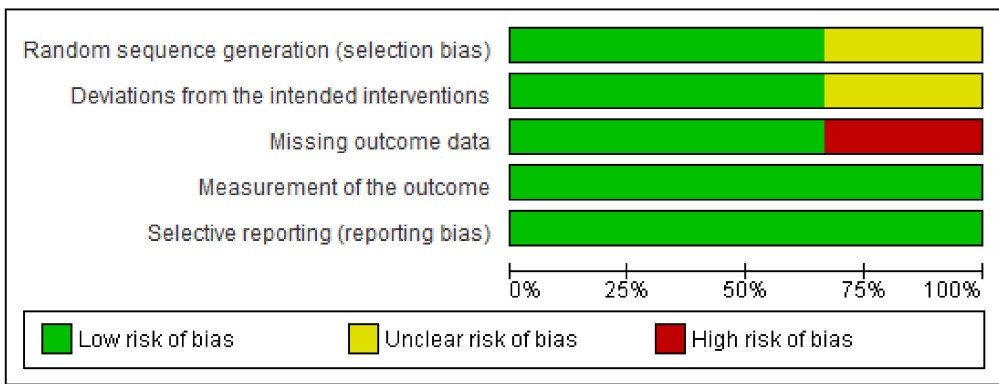

**Figure 3.** Risk of bias graph for randomised studies: review authors' judgements about each risk of bias item presented as percentages across all included studies.

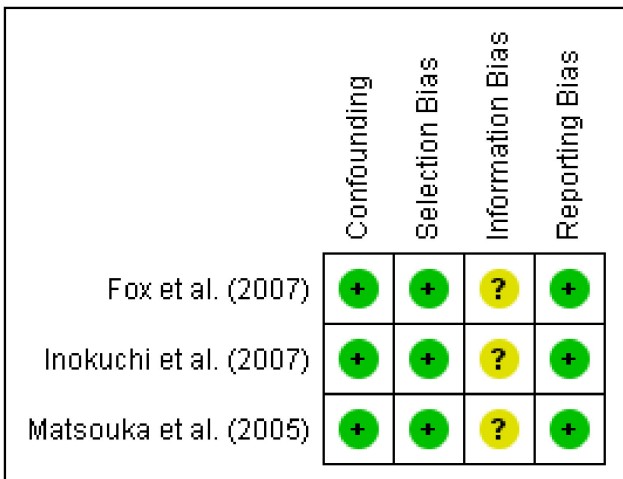

**Figure 4.** Risk of bias summary for non-randomised studies: review authors' judgements about each risk of bias item for each included study. (Fox, et al., (2007) [28]; Inokuchi, et al., (2007) [27]; Matsouka, et al., (2005) [25].

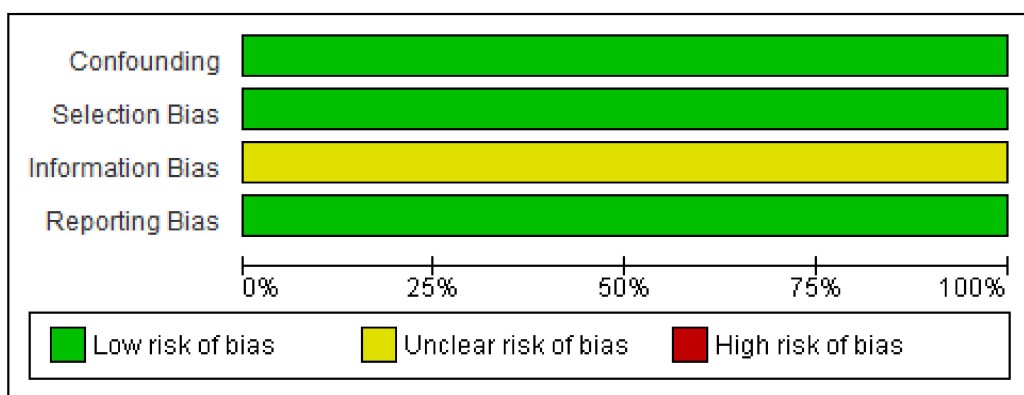

**Figure 5.** Risk of bias graph for non-randomised studies: review authors' judgements about each risk of bias item presented as percentages across all included studies.

## 4. Discussion

The overall findings of this systematic review demonstrate the variety of ways in which enjoyment can be targeted in group-based physical activity for older adults. While enjoyment outcomes (*n* = 3) appeared to improve, there was wide heterogeneity in the measurement tools. Enjoyable group-based physical activity for older adults may consist

of a supportive instructor and supportive peers, creating a shared positive experience. Components included confidence building through competence and courage experienced in the company of others. Within a physically supportive environment, older adults have the potential to generate social support to enjoy being physically active.

### 4.1. Features of Enjoyable Group-Based Exercise

The identified components of enjoyable group-based physical activity programmes for older adults are consistent with components associated with successful interventions to change physical activity [32]. These components aim to promote changes using established BCTs, engaging social support, delivered by a wide range of people with appropriate training, in a range of settings, with group and individual or mixed formats, focusing on maintenance and maximising frequency of contacts [32].

The overarching affective theme of courage may be an enabler to promote changes, and the use of BCTs, particularly the engagement of social support, facilitated individually through competence, interpersonally through courage, socially through company, and environmentally with opportunities to interact. The range of facilitators and settings and formats across the studies was evident, with 50% of studies having a longer-term duration of >6 months. The importance of meeting more than once a week [25], sustainability in the delivery context by profession [27] or virtual format [29], with particular focus on mood [28], the composition of the group [22], and a maintenance phase [26] was incorporated in the included studies. The contrasting and overlapping themes of identifying as part of a group that exercises together or identifying as an exerciser who meets in a group is discussed further.

### 4.1.1. Identifying as Part of a Group That Exercises Together

Identifying as part of a group that exercises together emphasises the importance of social interaction and the use of recreational and external activities. The use of social theories, such as the self-categorization theory in the GOAL trial [21], align with this approach. Social cognitive theory has previously been used to explain the effect of physical activity on well-being and how affective states can be drawn upon [33]. When affective outcomes are prioritised over health-related outcomes, this is known to positively predict physical exercise intentions among older adults [34]. Our findings have demonstrated individuals building their group identity on shared experiences, through collaboration and emotional support (e.g., tea-breaks, messages, and social activities) outside the exercise group. The importance of valuing interaction with peers, companionship, and the encouragement from others has already been identified as major social influences on the physical activity environment [35,36]. The companionship provided can be a distraction to the exercise itself, and not perceiving the activity to be "exercise" [36].

The courage theme outlined in our results is consistent with "motivation building" collaboration behaviours: encouragement, challenging one another, fostering persistence, and celebrating success [37], identified to shape continued exercise adherence, enjoyment, and relatedness. The encouragement was described as verbal support and positive reinforcement, which was evident in the design of the internet-based intervention [29]. The positive external reinforcement occurred during exercising, making a distinction from exercising alone or activities outside the exercise time. Here, it is considered that the group environment can be optimised to support exercise behaviour. The finding of a qualitative preference to group exercise, yet a high compliance to home exercises [28], has been examined elsewhere. A "true group"—who has experienced team building—as opposed to a standard group, is superior to a home-based intervention, in terms of adherence and social interaction [38]. However, home-based interventions that include contact from others are more superior than home-based with no contact [38]. Furthermore, there may be more willingness to meet together as a group again, after trying to perform the exercises individually at home.

The personal challenge or competence aspect of enjoyment is something that has been identified as a predisposing factor to physical activity among older adults [39] connected to building confidence and self-efficacy [37]. Perhaps the effect of observing others accomplish similar exercise challenges further enables individuals to maintain the behaviour. This vicarious experience is attributed to social role models who can raise one's self-efficacy [40] and increasing social interactions [41]. Additionally, the ability to be able to challenge someone appears to be built on having established rapport and being comfortable with others, and therefore willingly accepting the challenge. While it has been identified to be difficult to isolate the effect of specific BCTs, moderator analyses from older adult physical activity interventions lasting 12 months or more suggested that the BCT of feedback was related to more effective interventions [42]. Perhaps it is the desire for competence that makes participants more open to feedback. This practical intragroup coaching—exercise task-focused support—similar to other motivational building affective forms of support, arises out of the shared exercise experience.

### 4.1.2. Identifying as an Exerciser Who Meets in a Group

Identifying as an exerciser who meets in a group demonstrates the individual competence required to participate in group-based exercise. The use of SDT fits this approach in the ACE study [26], where individuals met with an activator who introduced them to existing groups. They were only brought together as an intervention group twice to share knowledge about existing opportunities and celebrate success as a motivation to maintain improved behaviour. This is an example of social support where individuals were sharing resources, but not necessarily experiencing cohesion. Competence (technique, practice, and discussions about activity) enabled individuals to contribute to the group identity.

The enjoyment of the physical activity itself, or the combination with enjoyable and useful activities, has been described elsewhere as an individual preference. This may depend on the individuals' task-focus (physical activity) or group identity-focus (social) motivations. These elements were somewhat explored as psychological mediators in the GOAL trial [23]. Among the intervention groupings of older adults, an attraction to the groups' social activities and enjoyment was higher, who also perceived their group to increase in social dimensions of cohesiveness over time, compared to the mixed-age group. There was no difference in attraction to the group task activities, which were similar, regardless of age-specific classes; however, in older age groups, the task cohesion also increased over time. The case for social cohesion increasing over time [43] is not consistent with the findings from Baez, et al., (2017), which demonstrated a non-significant improvement in enjoyment in both online-group exercising and home exercising arms over time. However, the implications of the intervention group not being in person and the control group having telephone reviews from the instructor, affected the depth of cohesion in both groups. The facilitation of group cohesion, by identifying opportunities for interaction among participants, was identified by the ACE study as a future recommendation, expected to improve adherence rates, perceptions of well-being, and long-term maintenance of positive outcomes [26].

While some aspects of engaging classes are structural and organization factors, e.g., high-quality instructors [36], the shared responsibilities of creating an engaging environment are between the instructor, participants, and their chosen activities. Multi-component interventions should consider both the maintenance phase of the group and how the individual can maintain exercise beyond the group to target enjoyment in group-based physical activity interventions for a diverse range of older adults. Provisions for a supportive physical and social environment with the interlock of competence, company, confidence, and courage are recommended to create positive group-exercise experiences targeting enjoyment among older adults.

*4.2. Limitations of Evidence*

There were a lack of studies in the literature that explicitly targeted or measured enjoyment. As enjoyment itself is poorly defined, measurement scales are not well established. None of the tools used were validated for use in older adults and were not comparable. Physical activity data was insufficiently measured, with only one included study using accelerometers. Furthermore, the overall strength of the evidence was weak due to some missing data, controls with varying levels of baseline physical activity and exercise behaviours. A meta-analysis was not appropriate to produce meaningful results on enjoyment outcomes, due to the heterogeneity in design, intervention content, and outcome measures, and was not possible for exercise outcomes. Other qualitative studies may provide more of an insight into the experience of enjoyment among older adults attending group-based exercise programmes, which did not meet the inclusion criteria of this study.

*4.3. Implications for Practice, Policy and Future Research*

The overall results present features of programmes that could potentially be used to promote enjoyment in group-based physical activity programmes for older adults. The interventions that implicitly targeted enjoyment were found to meet together more than once a week, with considerations for a home-based or online element, tailoring progress while maintaining similarity and identity in the group, building both collaboration and social support.

There was a high proportion of females in the included studies (71.3% overall), limiting broader generalisability of the findings. Given that only one study was exclusive to females [25], there is a need for further research on the enjoyment of group-based physical activity among older males. One exception recruited 66% males, based on their formative research to recruit more males, by focusing on the provision of assistance with getting out and about and engaging with their communities [26]. Their alternative study design focused on introducing older adults to a range of existing groups, rather than creating a new exercise group [26], and there is a need to explore alternative intervention designs that have a wider reach to under-represented groups in older adults physical activity interventions.

The findings suggest that interventions should target environmental, social, interpersonal, and individual components to create an enjoyable exercise experience for older adults (Table 3). There is a need for a further definition of the enjoyment of physical activity among older adults. Further research is needed on physical activity interventions explicitly targeting enjoyment among older adults, underpinned by types of affect, BCTs, and behaviour change theories.

**Table 3.** Environmental, social, interpersonal, and individual components of affect, BCTs, and behaviour change theory among physical activity interventions targeting enjoyment in older adults.

| | Affect | BCTs | Theory |
|---|---|---|---|
| Environmental | Welcoming, supportive (instructor), providing opportunities to interact—exercise programme or externally | Adding objects to the environment (exercise aids); restructuring the social environment (online, tea-breaks, activity opportunities) | Self-categorization theory (group cohesion) Self-determination theory (social support—competence) |
| Social | Company—shared experience, perceived similarity, building a sense of community | Social support—messaging, post-workout, and social events | |
| Interpersonal | Courage—supporting, interacting, caring, expressing (laughing) | | |
| Individual | Competence—performance, achievement, Confidence—self-worth, knowledge, satisfaction (noticing and valuing effort and benefits for daily life) | Instruction and demonstration on how to perform behaviour (training tips, manual, handout, videos) | |

## 5. Conclusions

To the authors' knowledge, this is the first systematic review to examine the effectiveness of group-based physical activity interventions targeting enjoyment in older adults. The findings suggest that multi-component interventions may be designed to increase older adults' enjoyment and adherence if they encourage individual competence and group cohesion simultaneously through confidence and company. The generalizability of the findings were limited by the under-representation of men in the included studies. Nonetheless, the findings have important implications for the design of future group-based physical activity programmes for older adults. Within a physically supportive environment, older adults have the potential to restructure their own social environment, generating social support to enjoy being physically active, through the creation of a welcoming environment as part of the development of an enjoyable exercise experience. However, the evidence is limited and an agreed definition of the meaning of enjoyable physical activity for older adults is required.

**Supplementary Materials:** The following supporting information can be downloaded at: https://www.mdpi.com/article/10.3390/jal2020011/s1, Figure S1: Search results from individual databases; Table S1: Ovid Medline Search strategy; Table S2: Examples of Behaviour Change Techniques commonly identified in group-based physical activity interventions targeting enjoyment in older adults.

**Author Contributions:** M.A.T. and R.M.C. developed the idea and protocol for the review. Searching, screening, data extraction and analyses were conducted by R.M.C. and M.A.T., with support from K.F.P. and N.E.B. The manuscript was drafted by R.M.C., with input from M.A.T., K.F.P. and N.E.B. All authors have read and agreed to the published version of the manuscript.

**Funding:** This research was funded by the Department for the Economy (DfE) Postgraduate Studentship (2020/2021—2022/2023). MAT is part funded by the NIHR (Public Health Research (NIHR131550)). The views expressed are those of the author(s) and not necessarily those of the NIHR or the Department of Health and Social Care.

**Institutional Review Board Statement:** Not applicable.

**Informed Consent Statement:** Not applicable.

**Data Availability Statement:** No new data were created or analysed in this study. Data sharing is not applicable to this article.

**Acknowledgments:** Support for accessing and searching databases, including subject headings and key words in developing the search strategy, was provided by Kelly McCoo, Assistant Subject Librarian—Life & Health Sciences, Ulster University.

**Conflicts of Interest:** The authors declare no conflict of interest. The funders had no role in the design of the study; in the collection, analyses, or interpretation of data; in the writing of the manuscript, or in the decision to publish the results.

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
