# Peer review of "Group-Based Physical Activity Interventions Targeting Enjoyment in Older Adults: A Systematic Review"

_2673-9259, doi:10.3390/jal2020011_

Round 1

Reviewer 1 Report

First, I would like to thank you for the opportunity to conduct the following review of the article entitled: “Group-based physical activity interventions targeting enjoyment in older adults: a systematic review”.Physical inactivity, which is often associated with aging, is one of the fundamental factors contributing to the development of frailty. Multi-component physical exercise programs are the most effective interventions to delay disability and other adverse events, but the adherence to these programs is essential to be effective. In this sense, detecting which factors increase enjoyment is fundamental to improve the benefits and to ensure adherence to physical exercise in the long term. The study is well written, which makes it easy to understand. Authors detail the theoretical framework on which they are based and argue about it in the discussion. However, there are some aspects that should be consider prior to publication:

Abstract:

There is a typing error in line 20.

Metodology:

Line 85: “Subject headings and key words (title and abstract) were used by authors to develop the search strategy”. Could the authors be more s explicit about which terms they have used?

Results:

Line 166. The authors mention “Nine articles from six studies were deemed eligible for inclusion following full-text 166 screening.”, but then they excluded 3 of them to conduct the review. I think that there should be an adequate explanation.

Line 267. “Adherence outcome measures” is a subtitle? It should be indicated.

Discussion:

Line 452. Although the authors' purpose is to synthesize the information previously presented, there should not be a table in the discussion.

Author Response

Reviewer Comments

Response (included text)

First, I would like to thank you for the opportunity to conduct the following review of the article entitled: “Group-based physical activity interventions targeting enjoyment in older adults: a systematic review”. Physical inactivity, which is often associated with aging, is one of the fundamental factors contributing to the development of frailty. Multi-component physical exercise programs are the most effective interventions to delay disability and other adverse events, but the adherence to these programs is essential to be effective. In this sense, detecting which factors increase enjoyment is fundamental to improve the benefits and to ensure adherence to physical exercise in the long term. The study is well written, which makes it easy to understand. Authors detail the theoretical framework on which they are based and argue about it in the discussion. However, there are some aspects that should be consider prior to publication:

We are grateful for the time you have taken to review this submission and pleased you found it understandable. We are also thankful for the helpful comments and requests for revisions, which we have addressed below.

Abstract

There is a typing error in line 20.

Thank you for pointing this out, the errant full stop has been deleted in line 20.

Methodology

Line 85: “Subject headings and key words (title and abstract) were used by authors to develop the search strategy”. Could the authors be more explicit about which terms they have used?

The search terms were included in the supplementary table. We believe this is a more effective way of presenting the exact search terms than only including a summary of them in the text. However, in light of the request, we have added the following sentence to the methods on lines  104-112: “Search terms included those relating to ‘older adult’ and ‘group-based’ and ‘physical activity’ and ‘affect’, and are detailed in Table S1 (Table S1: Search Strategy).”

Results

Line 166. The authors mention “Nine articles from six studies were deemed eligible for inclusion following full-text screening.”, but then they excluded 3 of them to conduct the review. I think that there should be an adequate explanation

Thank you for this comment. For clarity, we did not exclude any of the nine articles, but four of the nine articles were based on the same study (GOAL), so these were grouped to avoid double reporting. To aid readability, we have amended the text on lines 191–193 to: “Six interventions (from nine articles, where four articles related to the same intervention) were eligible for inclusion”. We trust this will prevent any future confusion.

Results

Line 267. “Adherence outcome measures” is a subtitle? It should be indicated.

Apologies for this error.  The subtitles in section 3.4.1. seem to have been lost on formatting. The following have now been amended:

3.4.1.1. Enjoyment outcome measures – line 284

3.4.1.2. Physical Activity outcome measures – line 294

3.4.1.3. Adherence outcome measures – line 304

Discussion

Line 452. Although the authors' purpose is to synthesize the information previously presented, there should not be a table in the discussion.

To our knowledge there is no restriction on including a table in the discussion. We believe this is a helpful table to summarise the key learning in an easily digestible format to supplement the written text, making the learning more readily adoptable for future research and practice. For this reason, we have moved the table to section 4.3. (line 518 – 538) and swapped the first row and first column table headings to improve formatting. We are happy to be guided by the Editor as to what the most appropriate format for the discussion should be for the journal.

Reviewer 2 Report

I applaud the authors for this work. Group-based physical activity interventions targeting enjoy3 ment in older adults: a systematic review.

•    Please make sure that the structure for citing published literature in the text, as well as the style of references in the References section, are consistent with the journal's style (see Instructions to Authors).
•    English language needs revision for style and syntax.
•    Abstract must be rewritten. 
•       I suggest to organize the introduction section. Please organize ideas. 

Author Response

Reviewer Comments

Response (included text)

I applaud the authors for this work. Group-based physical activity interventions targeting enjoyment in older adults: a systematic review.

Thank you for this supportive comment and the helpful suggested amendments.

Please make sure that the structure for citing published literature in the text, as well as the style of references in the References section, are consistent with the journal's style (see Instructions to Authors).

Thank you for pointing this out. The references have been amended to the journal instructions for citations in the text and reference list.

English language needs revision for style and syntax.

Thank you for this comment. We are unclear what specific aspects of the text need to be revised. We have re-read the document to try to identify areas for improvement. Any remaining issues will be picked up with the publisher in preparing the manuscript for submission.

Abstract must be rewritten. 

The abstract has been prepared in accordance with the journal’s instructions for authors. We have amended the text to improve the presentation.  

I suggest to organize the introduction section. Please organize ideas. 

Thank you for this suggestion. To make the organisation of the introduction easier to follow, we have added the following subheadings.

1.1.  Physical activity for older adults

1.2 Enjoyment as a predictor of physical activity

1.3 Behaviour change and maintenance

1.4. Aim of this review

Reviewer 3 Report

Thank you for the opportunity to review an interesting topic „Group-based physical activity interventions targeting enjoyment in older adults: a systematic review“.

This manuscript was written correctly. However, the References were not organized according to the recommendations of MDPI and JAL.

Nonetheless, I would like the Authors to reveal what new has been found. What is the novelty and relevance of this systematic review? Could the data be discussed more clearly in order to indentify the benefits of social support and to correlate this impact of physical activity on enjoyment, mental health or quality of life in a complex way?

Kind Regards

Author Response

Reviewer Comments

Response (included text)

Thank you for the opportunity to review an interesting topic„ Group-based physical activity interventions targeting enjoyment in older adults: a systematic review“

Thank you for this supportive comment. We appreciate the time you have taken to provide this review.

This manuscript was written correctly. However, the References were not organized according to the recommendations of MDPI and JAL.

Thank you for identifying this. The references have been amended to the journal instructions for citations in the text and reference list.

Nonetheless, I would like the Authors to reveal what new has been found. What is the novelty and relevance of this systematic review?

Thank you for this comment. We had felt that this was covered in the discussion. In light of your comment we have added additional text to the conclusion to ensure the concluding remarks properly highlight the novelty and relevance findings of this review.

Could the data be discussed more clearly in order to identify the benefits of social support and to correlate this impact of physical activity on enjoyment, mental health or quality of life in a complex way?

Thank you for this interesting comment. Whilst we agree that the relationship between these factors is important, the aim of the review was not to unpick them and consequently the data from the included studies did not explore the interaction between activity and mental health or quality of life. Our summary of the results indicate that social support was at least in part related to the enjoyment of the physical activity programme. This, and other related aspects, are discussed in section 4.1. We have added  further subheadings to help orientate the reader to the individual aspects of enjoyable physical activity. We trust this helps.

Reviewer 4 Report

Dear Authors:

Thank you very much for giving me the opportunity to review the article entitled: “Group-based physical activity interventions targeting enjoy-2 ment in older adults: a systematic review”.

The article is very interesting, it is well written and an adequate methodological analysis has been carried out. However, I note that a more detailed and comparative analysis of physical activity programs is lacking, indicating which activities are carried out in the warm-up, main part and cool-down. As well as the duration of each of them.

Author Response

Reviewer Comments

Response (included text)

Thank you very much for giving me the opportunity to review the article entitled: “Group-based physical activity interventions targeting enjoyment in older adults: a systematic review”.I note that a more detailed and comparative analysis of physical activity programs is lacking, indicating which activities are carried out in the warm-up, main part and cool-down. As well as the duration of each of them.

One of the objectives of this review was to identify the components that were specifically related to creating an enjoyable experience. We included a summary of the intervention content in Table 2. Whilst further details, such as specific activities carried out in the aspects of the exercise class would be interesting, the authors did not report their interventions at this level of detail.

Reviewer 5 Report

Dear authors,

Your submitted paper “Group-based physical activity interventions targeting enjoyment in older adults: a systematic review” is beneficial for an in-depth view of this issue.

Introduction

Row 32 – ageing; correct is aging

Row 40, 43, 54, 58, 75 and further in text – behaviour; correct is behavior

Row 68 – synthesise; correct is synthesize

Comments

All the necessary steps for conducting the systematic review were carefully followed. The sample of the chosen studies was composed of 73% females. However, your conclusions are recommended for all older adults??

Author Response

Reviewer Comments

Response (included text)

Your submitted paper “Group-based physical activity interventions targeting enjoyment in older adults: a systematic review” is beneficial for an in-depth view of this issue.

Thank you for this supportive comment.

Introduction

Row 32 – ageing; correct is aging

Thank you. ‘Aging’ is the American English spelling of the word. We have chosen ‘ageing’ (UK English spelling) as it is consistent with the title of the journal.

Row 40, 43, 54, 58, 75 and further in text – behaviour; correct is behavior

This also refers to the American English spelling. For consistency, we have used the UK English spelling.

Row 68 – synthesise; correct is synthesize

Again, this refers to the American English spelling. For consistency, we have used the UK English spelling.

Comments

 The sample of the chosen studies was composed of 73% females. However, your conclusions are recommended for all older adults??

Thanks, we readily acknowledge that this is a limitation of the review and added it to the limitations section (line 515-525): “There was a high proportion of females in the included studies (71.3% overall), limiting broader generalisability of the findings. Given that only one study was exclusive to females [25], there is a need for further research into enjoyment of group-based physical activity among older males. One exception recruited 66% males, based on their formative research to recruit more males, by focusing on the provision of assistance with getting out and about and engaging with their communities [26]. Their alternative study design focused on introducing older adults to a range of existing groups, rather than creating a new exercise group [26], and there is a need to explore alternative intervention designs that have a wider reach to under-represented groups in older adults physical activity interventions.”

In addition, we have amended the conclusions to acknowledge that this limitation of the generalisability of the findings (line 535-536): “The generalizability of the findings are limited by the under-representation of men in the included studies”.